# Antibody–Drug Conjugates in Breast Cancer: Current Status and Future Directions

**DOI:** 10.3390/ijms241813726

**Published:** 2023-09-06

**Authors:** Cynthia Mark, Jin Sun Lee, Xiaojiang Cui, Yuan Yuan

**Affiliations:** 1Department of Medicine, Samuel Oschin Comprehensive Cancer Institute, Cedars-Sinai Medical Center, Los Angeles, CA 90048, USA; 2Department of Surgery, Samuel Oschin Comprehensive Cancer Institute, Cedars-Sinai Medical Center, Los Angeles, CA 90048, USA

**Keywords:** antibody–drug conjugates, breast cancer, drug resistance

## Abstract

Antibody drug conjugates (ADCs) are novel medications that combine monoclonal antibodies with cytotoxic payloads, enabling the selective delivery of potent drugs to cancer cells expressing specific surface antigens. This targeted strategy seeks to optimize treatment effectiveness while reducing the risk of systemic toxicity, distinguishing ADCs from conventional chemotherapy. The rapid growth in ADC research has led to numerous developments and approvals for cancer treatment, with significant impacts on the management of breast cancer. ADCs like T-DXd for HER2-low disease and sacituzumab govitecan for triple negative breast cancer (TNBC) have provided valuable options for challenging subtypes of breast cancer. However, essential questions still need to be addressed, including the optimal order of ADCs amidst the growing number of newly developed ones and strategies to overcome resistance mechanisms. Preclinical studies have shed light on potential resistance mechanisms, emphasizing the potential benefit of combinational approaches with other agents such as immune checkpoint inhibitors (ICIs) and targeted tyrosine kinase inhibitors (TKIs) to enhance treatment effectiveness. Additionally, personalized approaches based on molecular profiling hold promise in tailoring ADC treatments to individual tumors, identifying unique molecular markers for each patient to optimize treatment efficacy while minimizing side effects.

## 1. Introduction

Antibody drug conjugates (ADCs) are a class of targeted medications that combine monoclonal antibodies with cytotoxic payloads [1]. In the field of oncology, ADCs are designed to selectively deliver potent cytotoxic drugs to cancer cells expressing specific surface antigens, such as HER2 and TROP-2. This targeted approach aims to maximize efficacy while minimizing the systemic toxicity associated with traditional chemotherapy. Given the vast potential of these drugs, this area quickly became an exciting field of research and has seen the number of ADCs developed and approved for use in cancer treatments increase rapidly in the last decade.

The mechanism of action of ADCs are quite complex and they work in a multifactorial manner (Figure 1). The ADC consists of three main components: a monoclonal antibody (mAb), a cleaver or non-cleavable linker, and a cytotoxic payload. The mAb component of the ADC is engineered to recognize and bind to a specific antigen that is overexpressed on the surface of cancer cells. Once the ADC binds to the target antigen, the entire complex is internalized into the cancer cells through receptor-mediated endocytosis. While receptor-mediated endocytosis is highly selective and requires specific receptors for internalization, pinocytosis is a more generalized process. In cases where cancer cells lack the target antigen for ADC binding, pinocytosis can still facilitate the uptake of ADCs, but conjugation to a large hydrophilic antibody limits the non-specific pinocytosis-mediated uptake in antigen-negative cells, thereby widening the therapeutic index and improving the overall safety of ADC treatment [2]. Within the cancer cells, the ADC is transported to the endosomes and subsequently fuses with lysosomes. This process triggers enzymatic degradation of the linker, leading to the release of the cytotoxic payload from the mAb. Once released from the linker, the cytotoxic payload exerts its cytotoxic effect within the cancer cells. The payload can be a wide range of chemotherapeutic agents, such as microtubule inhibitors or topoisomerase inhibitors. One of the unique features of ADCs is their potential to induce a bystander effect (also known as bystander killing). The bystander effect refers to the phenomenon where cytotoxic drugs released from ADCs can diffuse from targeted cancer cells to neighboring cancer cells that may not express the specific antigen recognized by the ADC. By leveraging the bystander effect, ADCs have the potential to target and eliminate both antigen-positive and antigen-negative cancer cells within the tumor, thereby improving treatment outcomes.

ADCs have emerged as a promising therapeutic strategy for various malignant diseases, including bladder cancer and ovarian cancer [3,4]. Notably, the field of breast cancer has witnessed significant advancements in the development of ADCs, with T-DM1 being the first approved ADC for solid tumors. Currently, two more ADCs (T-DXd and Sacituzumab govitecan) have received FDA approval for breast cancer. In this review paper, we aim to comprehensively examine the current landscape of ADCs utilized in breast cancer treatment, exploring their efficacy, potential resistance mechanisms, and future directions for further advancements in this field.

## 2. Antibody–Drug Conjugates in Breast Cancer

### 2.1. HER2-Targeting ADCs

HER2-targeting ADCs have emerged as a promising approach in breast cancer therapy, particularly for patients whose tumors express HER2. HER2 is a cell surface receptor that plays a crucial role in cell growth and proliferation. ADCs targeting HER2 typically utilize a mAb component that specifically recognizes and binds to HER2 receptors on the surface of cancer cells. Trastuzumab, a widely used HER2-targeting mAb, is frequently employed in HER2-targeting ADCs. The mAb component serves as a vehicle for selective delivery of a cytotoxic payload to HER2-positive cancer cells.

#### 2.1.1. Trastuzumab Emtansine (T-DM1)

The first ADC approved for the treatment of breast cancer was trastuzumab emtansine (T-DM1). T-DM1 is comprised of the anti-HER2 properties of trastuzumab and DM1, an maytansine derivative agent that works via microtubule inhibition, conjugated with a non-cleavable linker [5]. T-DM1 was shown to retain the mechanisms of action of trastuzumab, including inhibition of the PI3K/AKT signaling pathway [6]. After binding to HER2, T-DM1 is internalized and degraded, leading to the release of lysine-maleimidomethyl cyclohexane-1-carboxylate (MCC)-DM1 [7]. In initial clinical trials, Verma et al., 2012, showed that T-DM1 significantly improved overall survival (OS) when compared to lapatinib plus capecitabine (30.9 months vs. 25.1 months; *p* < 0.001) [5]. This led to the drug becoming FDA approved in 2013 specifically for use in patients with HER2-positive, metastatic disease who previously received trastuzumab and a taxane, either separately or in combination, and either received prior therapy for metastatic disease or developed disease recurrence during or within six months of receiving adjuvant therapy [8]. Continued clinical research on T-DM1 was reported in the EMILIA trial, a randomized, open-label, phase III clinical trial. Similar to the work produced by Verma et al., 2012 [5], the EMILIA trial showed an increased median OS when compared to lapatinib plus capecitabine (29.9 months vs. 25.9 months) [9]. More recent clinical trials have evaluated T-DM1 for use in early breast cancer. The KATHERINE trial evaluated patients with HER2-positive breast cancer found to have residual invasive disease after neoadjuvant therapy. The trial showed increased rates of disease-free survival at three years in the T-DM1 group as compared to the trastuzumab group (88.3% vs. 77.0%, respectively, *p* < 0.001). This trial led to an expanded FDA approval to include early breast cancer in 2019 [10]. FDA approval now includes use of TDM-1 in adjuvant treatment of patients with HER-2-positive early breast cancer who have residual invasive disease after neoadjuvant taxane and trastuzumab-based treatment [8].

#### 2.1.2. Trastuzumab Deruxtecan (T-DXd)

Similar to T-DM1, trastuzumab deruxtecan (T-DXd) is another ADC that consists of an anti-HER2 antibody linked to a cytotoxic agent. However, T-DXd contains a different payload, which is an exatecan derivative (DX-8951f derivative) that functions as a topoisomerase I inhibitor. Another feature of T-DXd is that it contains a specialized tetrapeptide-based cleavable linker that allows for selective cleavage in cancerous cells. This limits the premature release of the cytotoxic agent and reduces toxic effects [11,12]. This specialized linker allows this drug to have a significantly higher drug to antibody ratio of approximately 8, which is more than double that achieved in T-DM1 (drug to antibody ratio of approximately 3.5) [11,12]. After initial studies showed T-DXd to be a potent antitumor drug with a favorable safety profile [11,12], the drug quickly moved into clinical trial testing. DESTINY-Breast 03 was a phase III clinical trial comparing the safety and efficacy of T-DXd. This trial showed significant improvement in progression-free survival (PFS) as compared to T-DM1, as well as a manageable toxicity, which led to the FDA approving of the agent as a second line therapy in HER+ metastatic breast cancer (MBC) [13]. Moreover, T-DXd received FDA approval in HER2-low breast cancer based on the results of the DESTINY-Breast 04 trial, which was a phase III clinical trial involving patients with HER2-low metastatic breast cancer who had already received two lines of prior chemotherapy [14]. In the trial, T-DXd was shown to have a significantly higher median PFS as compared to other chemotherapies chosen by the physician (10.1 months vs. 5.4 months, respectively, *p* < 0.001) and improved OS (23.9 months vs. 17.5 months, *p* = 0.003) [14]. HER2-low BC was defined by IHC 1+ or 2+ and lack of HER2 amplification on FISH testing, and the HER2-low population accounts for up to 45–55% of all breast cancers. The successful trial and subsequent approval of T-DXd not only introduced a new subtype of breast cancer known as HER2-low disease, but also brought about a revolutionary change in the field of breast cancer treatment [15].

#### 2.1.3. Disitamab Vedotin (RC48)

Disitamab vedotin (RC48) is another novel ADC that utilizes HER2 mAb that is linked to monomethyl auristain E (MMAE), a microtubule inhibitor, via a cleavable valine-citrulline linker. It has a drug antibody ratio (DAR) of four [16]. However, unlike the previous two drugs, the anti-HER2 antibody of disitamab vedotin is not trastuzumab, but the antibody disitamab [17]. Disitamab has been shown to have a better molecular affinity for HER2 as compared to trastuzumab. MMAE is a synthetic derivative of auristatin which has an anti-mitotic effect [18]. Disitamab vedotin was first approved in China for the treatment of locally advanced or metastatic gastric cancer that was HER2-overexpressing [19]. There are ongoing clinical trials evaluating RC48 for use in HER2-positive breast cancer. Given disitamab’s higher affinity for the HER2 receptor, one area of ongoing research is evaluating RC48 for use in populations that have HER2-low expression. Wang et al., 2021, compiled data on two previous studies that showed RC48 was effective in HER2-positive and HER2-low expressing populations [20]. In the HER2-low expressing subgroup, the tumor objective response rate (ORR) and median PFS (mPFS) were 39.6% (95% CI: 25.8%, 54.7%) and 5.7 months (95% CI: 4.1, 8.3), respectively [20]. The common treatment-related adverse events (TRAEs) were elevated AST (64.4%), ALT (59.3%), hypoesthesia (58.5%), and neutropenia (48.3%). Grease 3 adverse events (AEs) were neutropenia (16.9%), increased GGT (12.7%), and fatigue (11.9%) [20]. Currently, multiple ongoing trials are testing the efficacy of RC48 in HER2-low advanced breast cancer (NCT05831878, CT04400695), hormone receptor-positive (HR+) MBC (CT05904964), or in combination with penpulimab as neoadjuvant therapy in HER2+ BC (NCT05726175).

#### 2.1.4. ARX-788

ARX-788 is another novel HER2-targeted mAb that is linked to the cytotoxic payload AS269. ARX-788 uses a synthetic amino acids para-acetylphenylalanine (pAF) that is incorporated into a predetermined site on the heavy chain of the monoclonal anti-HER2 mAb and allows for conjugation of a diverse array of payloads. In the case of ARX-788, the pAF is used to link the mAB to the payload, AS269. ARX-788 demonstrated activities in HER2-positive, HER2-low, and T-DM1 resistant tumors in preclinical studies [21]. In a combined analysis of two phase I studies in HER2-positive solid tumors in the U.S. and Australia (ACE-Pan tumor-01) and in HER2-positive breast cancers in China (ACE-Breast-01), ARX-788 was well tolerated, with most adverse events (AEs) being grade 1 or 2 [22]. The most common grade >3 AEs were ocular AEs (5.7%) and pneumonitis (4.3%) in the ACE-Breast-01 trial; pneumonitis (2.9%) and fatigue (2.9%) in the ACE-Pan tumor-01 trial. In the 1.5 mg/kg cohort, ORR was 74% (14/19) and 67% (2/3) for ACE-Breast-01 and ACE-Pan tumor-01, respectively. The disease control rate (DCR) was 100%. Median duration of response (DOR) or mPFS has not been reached [22]. High stability of ARX-788 and low serum exposure of pAF-AS269 may underlie the low systemic toxicity, which differentiates it from other ADCs [22]. The phase III ACE-Breast-02 trial comparing ARX-788 with lapatinib combined with capecitabine in patients with HER2-positive advanced MBC is eagerly awaited. A global, phase II Study of ARX788 in HER2-positive MBC patients who were previously treated with T-DXd is currently ongoing (NCT04829604). ARX-788 alone or in combination with PD1 inhibitor, cemiplimab, as neoadjuvant therapy in HER2+ early stage BC, is being evaluated in the I-SPY2 trial (NCT01042379).

### 2.2. TROP-2-Targeting ADCs

Trophoblast cell-surface antigen 2 (TROP-2) is a cell surface receptor that is overexpressed in various cancers, including breast cancer. TROP-2 was shown to be expressed in approximately 80% of breast cancers and was found to be an unfavorable prognostic indicator [23,24]. It has been found to be expressed across all breast cancer subtypes [25]. Initial studies into the receptor have shown that it is involved in regulating the growth and invasion of tumor cells [26]. TROP-2-targeting ADCs utilize a mAb designed to specifically recognize and bind to TROP-2 receptors on cancer cells.

#### 2.2.1. Sacituzumab Govitecan

Sacituzumab govitecan (Trodelvy^TM^) comprises a humanized mAb called sacituzumab, which targets the TROP-2 receptor, linked to a potent chemotherapy drug called SN-38, a metabolite of irinotecan. Once inside the cancer cell, SN-38 exerts its cytotoxic effects by inhibiting topoisomerase I, leading to DNA damage and cell death. Clinical studies have shown promising results in patients with metastatic triple negative breast cancer (TNBC) or HR-positive HER2-negative MBC, with significant improvement in PFS and OS compared to standard chemotherapy. In the phase III ASCENT trial, the effectiveness of sacituzumab govitecan was compared to single-agent chemotherapy in patients with heavily pretreated metastatic TNBC [27]. The results showed that sacituzumab govitecan significantly improved both PFS (median of 5.6 months versus 1.7 months with chemotherapy) and OS (median of 12.1 months versus 6.7 months with chemotherapy). The treatment was associated with more frequent myelosuppression and diarrhea but showed promising results in extending survival in this challenging-to-treat cancer. Furthermore, sacituzumab govitecan demonstrated efficacy for HR-positive HER2-negative breast cancer in the phase III TROPiCS-02 trial [28], where sacituzumab govitecan was compared with physician’s choice chemotherapy in patients with endocrine-resistant, chemotherapy-treated HR+ HER2-negative advanced breast cancer. The study showed that sacituzumab govitecan significantly improved PFS compared to chemotherapy, with a 34% reduction in the risk of disease progression or death. The treatment demonstrated a manageable safety profile and offered a promising option for patients with heavily pretreated HR+ HER2-negative breast cancer.

#### 2.2.2. Datopotamab Deruxtecan (Dato-DXd)

Dato-DXd is an investigational ADC composed of a humanized anti-TROP2 IgG1 mAb attached to a topoisomerase I inhibitor payload, deruxtecan, through a stable tetrapeptide-based cleavable linker. It shares the same highly potent payload as T-DXd but targets cells that express TROP2. In preclinical models, Dato-DXd demonstrated specific binding to TROP2-expressing tumor cells, leading to intracellular trafficking and release of a potent DNA topoisomerase I inhibitor (DXd) [29]. This resulted in DNA damage and apoptosis, leading to significant antitumor activity and tumor regression in TROP2-expressing xenograft models, while showing acceptable safety profiles. These findings suggested that Dato-DXd could be a promising treatment option for patients with TROP2-expressing tumors in clinical settings. Preliminary results from the TROPION-PanTumor01 study showed promising response rates and a manageable safety profile in heavily pre-treated patients with metastatic TNBC who received Dato-DXd [30]. The phase III TROPION-Breast02 trial (NCT05374512) aims to compare the efficacy and safety of Dato-DXd versus investigator’s choice of chemotherapy (ICC) as a first-line treatment in patients with locally recurrent inoperable or metastatic TNBC who are not candidates for PD-1/PD-L1 inhibitor therapy. Approximately 600 patients will be randomized to receive either Dato-DXd or ICC, and the study will assess PFS and OS as primary endpoints, along with several secondary endpoints including objective response rate, duration of response, and safety. Moreover, the efficacy and safety of Dato-DXd will be evaluated in patients with inoperable or metastatic HR+/HER2− breast cancer who have received one or two prior lines of systemic chemotherapy in the same setting in an ongoing phase III study TROPION-Breast01 (NCT05104866).

### 2.3. HER3-Targeting ADCs

HER3 plays a key oncogenic role in breast cancer, being associated with poor prognosis and resistance to PI3K/AKT/mTOR inhibitors and endocrine therapy [31]. Patritumab deruxtecan (HER3-DXd) is an ADC comprised of a fully human anti-HER3 IgG1 mAb, patritumab, conjugated to a topoisomerase 1 inhibitor payload, deruxtecan, via a tetrapeptide-based cleavable linker that has shown promising efficacy in patients with HER3-expressing MBC. Krop et al. reported the safety and efficacy of HER3-DXd in a phase I/II study of U31402-A-J101 (NCT02980341; JapicCTI-163401) in patients with heavily pretreated HER3-expressing MBC. The most common AEs were neutropenia (39.6%), thrombocytopenia (30.8%), anemia (18.7%), and leukopenia (18.1%). Twelve patients (6.6%) experienced interstitial lung disease (ILD), including one grade 5 event. Overall response rates were 30% in HR+/HER2−, 22.6% in TNBC, and 42.9% in HER2+ MBC [32]. 

The phase II ICARUS-BREAST01 trial evaluated HER3-DXd in patients with HR+HER2− MBC not selected for HER3 expression, who progressed on multiple lines of treatments including CDK4/6 inhibitors and chemotherapy [33]. A partial response was observed in 16 patients (28.6%), with stable disease in 30 patients and progressive disease in 10 (N = 56 total). The most common any-grade adverse events (AEs) were nausea (76.8%) and fatigue (89.3%); fatigue was also the most frequent grade ≥3 AE (14.0%). One patient had grade 1 confirmed ILD.

In the SOLTI TOT-HER3 trial (NCT04610528 part A), a window-of-opportunity trial evaluating a single dose of HER3-DXd in patients with treatment-naive HR+/HER2− early breast cancer, HER3-DXd was associated with clinical response, increased immune infiltration, and suppression of proliferation [34]. In the SOLTI TOT-HER3 trial part B presented at ESMO 2023 [35], analysis of baseline and post-treatment paired samples showed a statistically significant change in tumor cellularity and tumor-infiltrating lymphocyte (CelTIL) score overall (*p* = 0.046). The overall response rate (ORR) was 32% (35% in TNBC and 30% in HR+/HER2−). There was no association between baseline ERBB3 levels and CelTIL change or ORR. HER3-DXd induced a high expression of immune-related genes and suppressed proliferation-related genes. 

Hamilton et al. recently reported the part A results of a three-part phase II study examining the efficacy of HER3-DXd across MBC subsets (NCT04699630) [36]. The ORR was 35% (95% CIs 23.1, 48.1) for all patients, and the clinical benefit rate (CBR) was 48% (95% CIs 35.2, 61.6). Activities were seen across a broad range of HER3 expression: patient with ≥75% HER3 expression had an ORR of 33% and CBR of 50%, patients with HER3 25–74% expression had an ORR of 46% and CBR of 54%. There were four patients with HER3 < 25% expression, limiting efficacy assessment. The median DOR was 10.0 months (95% CIs 5.5, NA). The most common AEs were nausea (50%), fatigue (45%), diarrhea (37%), vomiting (32%), and alopecia and anemia (30% each). Seven patients (12%) experienced a serious AE, including four patients (7%) with interstitial lung disease, nausea/vomiting, pneumonitis, and thrombocytopenia. This data confirmed the clinical activity in heavily pretreated MBC across the broad range of HER3 expression levels.

### 2.4. LIV1 Targeting ADCs

LIV-1 is a transmembrane protein with zinc transporter and metalloproteinase activity, which is associated with epithelial mesenchymal transition (EMT) with moderate/high level of expression in the majority of breast cancers [37]. Ladiratuzumab vedotin is a LIV-1 directed ADC that is composed of a mAb of LIV-1 conjugated to microtubule-disrupting agent, the MMAE payload, through a protease cleavable linker. Ladiratuzumab vedotin is currently being studied in multiple early phase clinical trials on breast cancer patients, as a monotherapy or in combination with other agents, including immune checkpoint inhibitors (ICIs), with encouraging clinical activities [38]. In second-line refractory triple negative breast cancer patients, ladiratuzumab vedotin at a dose of 1.25 mg/kg showed an ORR of 28%, confirming the promising activity of this agent [39]. Treatment related AEs were nausea (60%), fatigue (58%), neuropathy (54%), decreased appetite (44%), and constipation (39%). The most common ≥ grade 3 treatment AEs were neutropenia (21%), fatigue (14%), and hyperglycemia, hypokalemia, and hypophosphatemia (12% each). The combination of ICIs plus ladiratuzumab vedotin is under evaluation in an ongoing open-label phase Ib/II trial (SGNLVA-002/KEYNOTE 721) assessing the ICI pembrolizumab plus ladiratuzumab vedotin as a first line therapy in patients with advanced TNBC [40]. The combination achieved an ORR of 35% (N = 66), including 2 cases of complete response and 21 partial responses. Stable disease was achieved in 32 subjects; diarrhea, nausea, fatigue, peripheral neuropathy, and neutropenia were the most observed treatment-related toxicities [40].

## 3. Challenges and Resistance Mechanisms

With the existing availability of multiple ADCs and anticipated arrival of newer ADCs in breast cancer, new challenges and unmet clinical needs arise, which include a better understanding of the mechanisms of resistances and strategies for optimal ordering of ADCs [41]. For example, with the recent approvals of sacituzumab govitecan for HR+/HER2− and metastatic TNBC as well as T-DXd for HER2-low MBC, many patients are now candidates for multiple ADCs. However, given potential cross-resistance based on antibody target versus payload [42], the optimal order remains uncertain.

Abelman et al. evaluated the safety and efficacy of another ADC after treatment with an ADC for patients with MBC [41]. In this study, 32 patients were identified as having received more than one ADC. The median PFS on the first ADC used (ADC1) was significantly longer, at 7.55 months compared to a median of 2.53 months on the second ADC (ADC2) (*p* = 0.006). PFS for ADC2 with antibody target change was 3.25 months compared to 2.30 months with no target change (*p* = 0.16). When the second ADC contained the same antibody target as the first, cross-resistance was present in 9/13 cases (69.2%), compared to 8/16 cases (50.0%) when the second ADC targeted a different tumor antigen. This result suggests a probability of cross-resistance to ADC after ADC treatment, while others had durable responses on latter lines of therapy, particularly if the antibody target was switched. Further research is needed for validation and optimal selection of ADC-based treatment options.

### 3.1. Bystander Effect (Bystander Killing) and Toxicity

ADCs were designed to affect target cells expressing a particular target antigen; however, premature release of the cytotoxic payload can lead to significant toxicity. The ideal target antigen is one that is homogenously expressed on the surface of tumor cell, but it can be challenging to identify a receptor that is homogenously expressed on cancerous cells and not expressed on healthy cells [43]. One workaround for cancers that may express target antigens in a heterogeneous way is the use of something called the bystander effect (also called bystander killing). In this case, a cytotoxic payload is designed so that it is membrane-permeable and can enter the extracellular spaces around the target cell and facilitate the death of neighboring cells. This can also lead to damage of supporting structures, such as blood vessels or stromal cells, and can aid in destruction of the tumor [44]. However, given the leakage of the cytotoxic payload outside of the target cell, this also makes the drug more likely to kill healthy, non-cancerous cells and lead to toxic effects. The effect of the bystander effect can be either enhanced or diminished [45]. This bystander potency is determined based on the charge of the linker-drug derivative released from ADCs [1].

### 3.2. Tumor Heterogeneity

Breast cancer is a heterogeneous disease caused by a diverse population of cells with varying gene expression profiles. Heterogeneity within a tumor increases its ability to adapt to constantly changing constraints, but adversely affects a patient’s prognosis, treatment response, and clinical outcome. Intratumoral heterogeneity results from a combination of extrinsic factors from the tumor microenvironment and intrinsic parameters from the cancer cells themselves, including their genetic, epigenetic, and transcriptomic traits, their ability to proliferate, migrate, and invade, and their stemness and plasticity attributes [46]. Cell plasticity constitutes the ability of cancer cells to rapidly reprogram their gene expression repertoire, to change their behavior and identities, and to adapt to microenvironmental cues. These features also directly contribute to tumor heterogeneity and are critical for tumor progression. Breast tumor heterogeneity is one of the major factors contributing to drug resistance, recurrence, and metastasis after systemic treatments. ADCs targeting surface cancer cells expressing antigens often suffer from issues associated with tumor heterogeneity. For example, intratumor heterogeneity of HER2 expression was observed in patients with HER2-positive breast tumors [47]. Tumor heterogeneity represents an obstacle for achieving truly effective treatment using mAb targeting cancer cells. 

### 3.3. Resistance Mechanisms

#### 3.3.1. Receptor Modulation

Modulation of receptor expression has been shown to be one of the resistance mechanisms to ADCs through downregulation of target antigens. HER2 downregulation is a prime example of this. Multiple studies have found that the primary resistance mechanism for patients treated with T-DM1 was downregulation of the HER2 receptor [48,49]. It has been shown that HER2 positivity can change during treatment, and there are several documented studies that show that loss of the HER2 receptor is associated with worse outcomes [50,51,52]. Unsurprisingly, patients who have loss of HER2 have been shown to have poorer PFS as compared to patients who retain HER2 positivity when treated with T-DM1 (1.5 months vs. 6.0 months, respectively, *p* = 0.01) [53]. The SePHER study also showed that T-DM1 had decreased efficacy in patients that were previously treated with pertuzumab and that patients treated with pertuzumab had biopsies that confirmed downregulation of the HER2 receptor [54].

Similar to HER2 expression, a lowered expression of TROP-2 has been shown to be linked to resistance to sacituzumab govitecan. In one study, genomic analysis of a patient with triple negative breast cancer that was exhibiting progression on sacituzumab govitecan showed that the patient lacked TROP-2 expression [42]. It has also been shown that patients with lower TROP-2 expression have reduced mPFS when treated with sacituzumab govitecan as compared to those with higher expression (2.7 months vs. 6.9 months, respectively) [55]. Zhu et al., 2022, evaluated how various treatments could alter TROP-2 expression and found that treatment with tamoxifen significantly increased TROP-2 expression [56].

Another type of modulation that can be considered a resistance mechanism is alteration of the receptor itself. Previous research had already found variations in the HER2 receptor, called p95HER2, that have been associated with trastuzumab resistance [57]. Expression of p95HER2 has been shown to occur in approximately 30% of HER2-positive breast cancers [58,59]. The p95HER2 mutation has been shown to be correlated with trastuzumab resistance [60]. This mutation should be considered as another possible cause of resistance for anti-HER2 ADCs.

#### 3.3.2. Alterations in Internalization and Lysosomal Function

Another well-established resistance pathway is alterations that lead to defects in the internalization process. ADCs often require the cell to internalize the drug to lead to release the payload, and without that internalization process the drug becomes significantly less effective. The most commonly used pathway for drug internalization utilized by ADCs is through a clathrin-mediated pathway [61]. Although cells can also utilize a caveolae-mediated or a clathrin-caveolin-independent endocytosis for internalization [61,62]. Alterations in these mechanisms can change trafficking to the lysosome. Sung et al., 2017, developed a line of HER2+, T-DM1 resistant cells called N87-TM that were studied for resistance mechanisms. They found that the N87-TM cells internalize T-DM1 into caveolin-1 (CAV1)-positive puncta, which correlated to reduced response to T-DM1 [49]. They suggest that evaluation for caveolae-mediated endocytosis could serve as a novel predictor for response to T-DM1.

Additional mechanisms of resistance involve alterations in membrane transporters. These transporters can be located on the cell membrane or the membranes of lysosomes. Membrane transporters are utilized by cells to transport various molecules across that membrane. Upregulation or inhibition of these transporters can have a drastic impact on the effectiveness of chemotherapy. Oftentimes, these transporters are upregulated to increase chemotherapy efflux out of cells, leading to cytoprotection of the cell [63]. One of the best-studied membrane transporters is Multidrug Resistance 1 (MDR1). MDR1 is initially expressed on the cell membrane and becomes endocytosed when lysosomes are formed. MDR1 is present on the membrane of the lysosomes and can mediate what is transported in and out of the lysosome. Tumors with high expression of MDR1 have been shown to have resistance to MMAE, a cytotoxic drug used in a number of ADCs [63]. Li et al., 2018, demonstrated that breast cancer cells can develop resistance to T-DM1 via multiple pathways, including decreased HER2 receptor expression and upregulation of MDR1 [64]. They additionally show that resistance to T-DM1 can be overcome with inhibition of MDR1.

## 4. Strategies to Overcome Resistance Mechanism

### 4.1. Combination Strategies

#### 4.1.1. Immunotherapy Combination

The combination of ADCs with immunotherapy offers a strong biological rationale, as ADCs can trigger mechanisms like immunogenic cell death and T cell infiltration, while immune-checkpoint inhibitors reinvigorate exhausted T cells, leading to potential synergistic effects [65]. Encouraging signals from early phase clinical trials support the ongoing investigation of combination regimens across multiple tumor types to enhance patient responses and overcome resistance. The BEGONIA trial is an ongoing study evaluating novel combinations of immunotherapies, including durvalumab, as first-line therapies for advanced or metastatic TNBC. The preliminary results of Dato-DXd and durvalumab combination among the 29 patients showed an ORR of 74%, with 7% achieving complete responses and 67% achieving partial responses, and the treatment was well-tolerated with manageable side effects [66]. The KATE2 study aimed to test the combination of atezolizumab and T-DM1 in patients with HER2-positive advanced breast cancer that had progressed after previous treatment with trastuzumab and a taxane [67]. However, the trial did not show a significant improvement in PFS, and patients receiving the combination experienced more adverse events compared to those on a placebo. In the study, a potential survival benefit was observed in the subgroup of patients with positive PD-L1 expression. This suggests that further investigation may be beneficial, specifically in the subset of patients with PD-L1-positive breast cancer. Currently, there are several ongoing trials evaluating the effectiveness of ADCs combined with ICIs (Table 1), including a phase Ib/II Morpheus-panBC (NCT03424005) evaluating the role of ICI atezolizumab combined with ladiratuzumab vedotin or sacituzumab govitecan.

#### 4.1.2. Targeted Therapy Combination

Targeted agents combined with ADCs have been studied mainly in HER2+ breast cancer without great success. The phase III KAITLIN study aimed to improve the efficacy and reduce the toxicity of high-risk HER2-positive early breast cancer treatment by replacing taxanes and trastuzumab with T-DM1 [68]; however, the primary endpoint was not met, and both arms achieved favorable invasive disease-free survival, with trastuzumab plus pertuzumab plus chemotherapy remaining the standard of care for high-risk HER2-positive early breast cancer. The phase III KRISTINE trial compared neoadjuvant treatments for HER2-positive breast cancer, showing that traditional chemotherapy plus dual HER2-targeted blockade (trastuzumab and pertuzumab) achieved a significantly higher pathological complete response rate than T-DM1 plus pertuzumab, though the latter had fewer grade 3–4 and serious adverse events [69]. Given that T-DM1 has less efficacy than newer ADCs, ongoing clinical trials are adopting new ADCs combined with HER2-targeted treatment, including TKIs (Table 1).

### 4.2. Bispecific Antibody–Drug Conjugates

Bispecific antibody–drug conjugates (BsADCs) represent a promising candidate in the field of targeted cancer therapy. These innovative molecules combine the advantages of traditional ADCs with the potential to engage dual tumor-associated antigens or dual epitopes, thereby enhancing tumor targeting and treatment efficacy [70]. BsADCs offer several key advantages over traditional ADCs. By targeting two distinct antigens simultaneously, these compounds enhance tumor cell specificity and reduce toxicity to normal tissues. Additionally, the dual-targeting approach can overcome drug resistance caused by decreased single-target expression, offering the potential for improved treatment outcomes in patients with resistant cancers. Zanidatamab zovodotin (ZW49), a bispecific anti-HER2 IgG1 antibody conjugated to a microtubule inhibitor auristatin payload (ZD02044) via a protease cleavable linker, is under development for HER2 expressing solid tumors (NCT03821233). Despite the promises of BsADCs, identifying ideal candidate antibody molecules for ADC drugs is a complex process. The limited availability of targets in breast cancer, such as HER2 and HER3, restricts the rapid construction of these molecules. Thus, comprehensive screening and validation of potential antibodies are essential to ensure successful BsADC development.

### 4.3. Dual Payloads

Bispecific antibody breast cancer often consists of distinct cell populations, each with unique gene expression profiles. To address this heterogeneity in cell populations, the development of ADCs with dual payloads has been pursued. For instance, a fibroblast growth factor 2 (FGF2)-conjugate bearing two cytotoxic drugs with independent modes of action, namely α-amanitin and monomethyl auristatin E, has been reported [71]. Yamazaki et al. demonstrated the therapeutic potential of these dual-drug ADCs in preclinical models [72]. However, further investigation is required to fully understand the potential of this molecular format.

## 5. Conclusions

ADCs have emerged as a revolutionary approach in cancer therapeutics, offering a novel way to deliver cytotoxic medication with reduced toxicity to healthy tissues. Following the approval of the first ADC for breast cancer, numerous ADCs have been in development. The approval of T-DXd for HER2-low disease has significantly impacted the landscape of breast cancer treatment, and SC for TNBC provides a valuable option for a challenging-to-treat subtype. However, several critical questions remain to be addressed, including the ideal choice of ADCs in the context of the growing number of newly developed ADCs, as well as the development of strategies to overcome resistance mechanisms. Preclinical studies have provided insights into potential resistance mechanisms to ADC therapy, highlighting the need for combinatorial approaches with other agents such as ICIs and targeted TKIs to enhance treatment efficacy. Additionally, personalized approaches based on molecular profiling of individual cancer cells are expected to play an instrumental role in improving cancer patient outcomes by tailoring ADC treatment strategies to the specific characteristics of each tumor. By identifying specific molecular markers unique to each patient, the most appropriate ADC or combination therapy can be selected to maximize treatment efficacy while minimizing potential side effects.

## Figures and Tables

**Figure 1 ijms-24-13726-f001:**
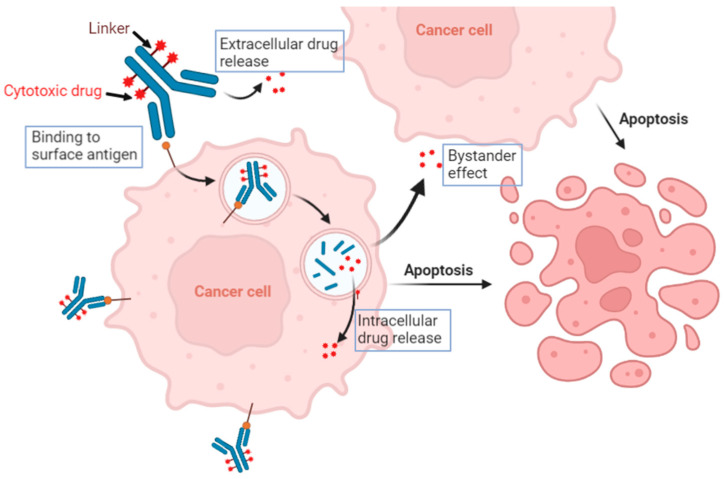
Graphic depicting the multifactorial mechanism of action of ADCs. ADCs are composed of three distinctive parts: the antibody, linker, and cytotoxic drug. Binding of the antibody to the antigen triggers intracellular cascades that can lead to internalization and release of the cytotoxic drug. ADCs can also have an effect on nearby cells in a process called the bystander effect. During this process, there is extracellular release of the cytotoxic drug leading to apoptosis of nearby cells.

**Table 1 ijms-24-13726-t001:** Novel ADCs Combination Trials.

ADC	ADC Target	Trials	Phase	Combination Therapy	Patient Population	Key Objective	NCT No.
Trastuzumab Emtansine (T-DM1)	HER2	KATE2	II	T-DM1+ atezolizumab	2L HER2+ MBC	PFS	NCT02924883
		CompassHER2 RD	III	T-DM1 + tucatinib	HER2+ stage II–III with residual disease after neoadjuvant treatment	iDFS	NCT04457596
Trastuzumab deruxtecan (T-DXd)	HER2	DESTINY Breast-08	IB/II	Capivasertib, Anastrozole, fulvestrant	1–2L HER-2 low MBC	AEs	NCT04556773
		TALENT	II	Anastrozole	Neoadjuvant HER2-low HR+	pCR	NCT04553770
		DASH	I	AZD6738	HER2+ advanced solid tumor	RP2D	NCT04704661
			I	Nivolumab	Advanced breast or urothelial Ca	DLT, ORR	NCT03523572
			IB	Pembrolizumab	Advanced breast cancer or NSCLC	DLT, ORR	NCT04042701
		DESTINY Breast-07	I/II	Durvalumab or pertuzumab, or paclitaxel +/− durvalumab or tucatinib	HER2+ MBC	AEs	NCT04538742
Disitamab vedotin (RC48-ADC)	HER2	ROSY	III	Endocrine therapy	1st line endocrine resistant HER2-low MBC	PFS	NCT05904964
			I	Penpulimab (AK105)	Neoadjuvant HER2-low BC	pCR	NCT05726175
ARX-788	HER2		II	Pyrotinib (HER2 TKI)	Neoadjuvant stage II–III HER2+ BC	RCB	NCT04983121
		ISPY-2.2	II	Cemiplimab	Neoadjuvant stage I–III HER2+ BC	pCR	NCT01042379
Sacituzumab Govitecan	TROP-2	ASCENT05	III	Pembrolizumab	Post-neoadjuvant stage I–III TNBC with residual disease	iDFS	NCT05633654
		ASCENT04	III	Pembrolizumab	1st line PD-L1+ metastatic TNBC	PFS	NCT05382286
		ASSET	I	Alpelisib	2^+^L HER2+ MBC	RP2D	NCT05143229
			I/II	Talazoparib	Metastatic TNBC	DLT	NCT04039230
			I	GS9716 (Mcl-1 antagonist)	Advanced solid tumors including MBC	DLT	NCT05006794
Dato-DXd	TROP-2	TROPION Breast 03	III	Durvalumab	Post-neoadjuvant stage I–III TNBC with residual disease	iDFS	NCT05629585
		ISPY-2.2	II	Durvalumab	Neoadjuvant stage I–III TNBC	pCR	NCT01042379
		PETRA	I/II	AZD5305 (PARPi)	Advanced solid tumor including breast cancer	DTL	NCT04644068
Ladiratuzumab vedotin (SGN-LIV1A)	LIV-1	MORPHEUS-panBC	II	Atezolizumab	Metastatic or Locally Advanced Breast Cancer	ORR	NCT03424005
		SGNLVA-002 Or KEYNOTE 721	I/II	Pembrolizumab	Advanced TNBC	ORR	NCT03310957
Patritumab Deruxtecan	HER3	VALENTINE	II	Endocrine therapy	Neoadjuvant high risk HR+/HER2− Early Stage BC	pCR	NCT05569811

PFS: progression-free survival; iDFS: invasive disease-free survival; AEs: adverse events; pCR: pathologic complete response; PR2D: recommended phase II dose; DLT: dose limiting toxicities; ORR: overall response rate; RCB: residual cancer burden.

## Data Availability

Not applicable.

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
