# Peer review of "Antibody–Drug Conjugates in Breast Cancer: Current Status and Future Directions"

_ijms, 2023, doi:10.3390/ijms241813726_

Round 1
Reviewer 1 Report
Dear Authors,
The manuscript (review) about Antibody Drug Conjugates (ADCs) is very organized and well-written. The writing style is fluid, clear, concise, and critical. The Authors exhibit an ample understanding of the literature, including some non-favorable aspects of using ADCs as therapeutic complexes, explaining the development of resistance mechanisms. The manuscript possesses sufficient merit to be published. However, some minor points should be addressed.
1. Sometimes, the P-value is annotated as capital "P" and minor case "p" in some other situations. Please, be consistent in the whole manuscript. A suggestion is to annotate capital and Italic "P" consistently.
2. Consider using Italic font for the word "via"; it should be "via."
3. Are there clinical trials using TROP-2-targeting ADCs? What is known about the mechanism of action of TROP-2-targeting ADCs? If the Authors have some answers to these questions should be included in the 2.2. TROP-2-targeting ADCs section.
4. From this reviewer's side, no further revision should be necessary.
Respectfully,
Academic Reviewer
MDPI Journals
Author Response
Dear reviewer,
Thank you for your comments. I made changes as below:
- All the 'P'-value was edited as your recommendation.
- All the 'via' was changed to Italic font.
- The clinical trials of TROP-2 targeting ADCs were already described in section 2.2.1 and 2.2.2 for each drug, as was MOC of each drugs.
I deeply appreciate your thoughtful review.
Thanks,
Reviewer 2 Report
This manuscript reviewed the ADCs for the BC treatment.
While the manuscript is well written, the following modifications would enhance the quality of the current paper.
[1] Given that BC treatment landscape and strategy have been considerably affected by Enhertu, it would be beneficial to have description and a summary table for FDA-approved ADCs for BC. Additionally, a summary of comparable table containing the essential reference, clinical study, safety profile, list of AEs, and dose regimens etc would be valuable
[2] Visual aid/schematic presentation and figures would help readers better understand the vast information for the BC treatment described in this manuscript.
[3] The authors should include the study's key objective listed in Table 1 for comparison in addition to the drug being used for the combination treatment. Otherwise, it's merely a list of clinical studies.
[4] It would be valuable to have the chemical structure of the linker and the payloads described in the text for readers who are not so familiar with the ADC field.
Minor
[1] lines 18, 294, 298 etc.
What do the authors mean by "sequencing"? Perhaps need to use an alternative word to replace “sequencing” and / or expand the description as I don't think the "sequencing" is an appropriate description for the context.
[2] Suggest using a more receptive and better-known trade name Trodelvy instead of SG abbreviation given by the authors.
[3] ] Were there any observed cross-resistance for the two ADCs that target the two different antigens but have different payloads vs. the same payloads?
- Authors should state HR is an abbreviation for hormone receptors, especially for those unfamiliar with the BC treatment.
Author Response
Dear Reviewer,
Thank you for your insightful review. Please see our response below:
- Two other FDA approved ADCs were named in the introduction section. The details of each drug was already described in the body of manuscript since there are only 3 medications.
- Key objectives were added in Table 1.
- The word ‘sequencing’ was replaced.
- ‘SG’ was spelled out as Sacituzumab govitecan to avoid using its trade name.
- There was no studies found for the two ADCs that target the two different antigens but have different payloads vs. the same payloads.
Thank you.